# Hepatitis B Surface Antigen Isoforms: Their Clinical Implications, Utilisation in Diagnosis, Prevention and New Antiviral Strategies

**DOI:** 10.3390/pathogens13010046

**Published:** 2024-01-03

**Authors:** Ivana Lazarevic, Ana Banko, Danijela Miljanovic, Maja Cupic

**Affiliations:** Institute of Microbiology and Immunology, Faculty of Medicine, University of Belgrade, 11000 Belgrade, Serbia; ana.banko@med.bg.ac.rs (A.B.); danijela.karalic@med.bg.ac.rs (D.M.); maja.cupic@med.bg.ac.rs (M.C.)

**Keywords:** hepatitis B virus (HBV), HBsAg isoforms, large surface protein (LHB), middle surface protein (MHB), small surface protein (SHB), mutation, diagnosis, new biomarkers, vaccine, new antivirals

## Abstract

The hepatitis B surface antigen (HBsAg) is a multifunctional glycoprotein composed of large (LHB), middle (MHB), and small (SHB) subunits. HBsAg isoforms have numerous biological functions during HBV infection—from initial and specific viral attachment to the hepatocytes to initiating chronic infection with their immunomodulatory properties. The genetic variability of HBsAg isoforms may play a role in several HBV-related liver phases and clinical manifestations, from occult hepatitis and viral reactivation upon immunosuppression to fulminant hepatitis and hepatocellular carcinoma (HCC). Their immunogenic properties make them a major target for developing HBV vaccines, and in recent years they have been recognised as valuable targets for new therapeutic approaches. Initial research has already shown promising results in utilising HBsAg isoforms instead of quantitative HBsAg for correctly evaluating chronic infection phases and predicting functional cures. The ratio between surface components was shown to indicate specific outcomes of HBV and HDV infections. Thus, besides traditional HBsAg detection and quantitation, HBsAg isoform quantitation can become a useful non-invasive biomarker for assessing chronically infected patients. This review summarises the current knowledge of HBsAg isoforms, their potential usefulness and aspects deserving further research.

## 1. Introduction

Infection with the hepatitis B virus (HBV) remains a global health problem despite successful vaccination programs. Almost two billion people worldwide have been infected with HBV at some point in their lives, and around 296 million live with chronic infection [1,2]. Complications related to chronic infection are still a source of significant morbidity and mortality of approximately 820,000 annually, primarily due to liver cirrhosis and hepatocellular carcinoma (HCC).

The natural history of chronic HBV infection is exceptionally complex and progresses nonlinearly through five distinct phases [3]: (1) hepatitis B e antigen (HBeAg)-positive chronic HBV infection, (2) HBeAg-positive chronic hepatitis B (CHB), (3) HBeAg-negative chronic HBV infection, (4) HBeAg-negative active CHB and (5) the HBsAg-negative phase. Patients in the last phase can experience viral reactivation in cases of severe immunosuppression. The current management of chronic HBV infection includes two therapeutic regimes: pegylated interferon α (PEG-IFNα) and nucleos(t)ide analogues (NAs). The treatment goal determined by all guidelines is the improvement of the long-term outcomes by the persistent inhibition of HBV replication [3,4,5]. Unfortunately, a complete, sterilising cure is not achievable by current therapy approaches because the suppression of viral replication does not affect the persistence of the HBV mini-chromosome—the covalently closed circular DNA (cccDNA) in the nucleus of the hepatocyte or the viral DNA integrated into the host genome. Thus, an alternative endpoint of therapy followed today is a “functional cure”, defined by a persistently undetectable serum HBV DNA and hepatitis B surface antigens (HBsAg) with or without seroconversion to corresponding antibodies [6].

The human HBV is the prototype member of the family *Hepadnaviridae*, which includes a variety of similar avian and mammalian viruses. The mature infectious HBV particle, originally called a “Dane particle”, consists of a nucleocapsid core enclosed in a glycolipid envelope. The nucleocapsid comprises partially double-stranded circular DNA attached to endogenous polymerase and an icosahedral capsid, while the envelope is made of a lipid bilayer bearing three different surface proteins [7].

The three viral surface proteins large (LHB), middle (MHB), and small (SHB) constitute the HBV surface antigen (HBsAg). It is a multifunctional glycoprotein and the major antigen of the viral envelope, responsible for eliciting humoral and cellular immune responses during infection. HBsAg isoforms are involved in many biological functions during HBV infection—from initial and specific viral attachment to the hepatocytes to establishing chronic infection with their immunomodulatory properties. The genetic variability of domains encoding HBsAg isoforms is responsible for their altered synthesis and presentation. Based on accumulated evidence, this may play a role in the pathogenesis of specific liver conditions. In diagnosis, the quantitation of HBsAg isoforms has been perceived as a potentially valuable tool for evaluating the chronic infection phase and monitoring to the treatment response. The immunogenic properties of surface proteins were long ago recognised as crucial for the development of vaccines, and the presence of different HBsAg isoforms has a role in the immune response they elicit. Some of the new therapeutic approaches that have been developed in recent years target the production, assembly and secretion of HBsAg isoforms. This review summarises the current knowledge of HBsAg isoforms, their potential usefulness and their aspects deserving further research.

## 2. HBV Molecular Virology and Genetic Variability

The organisation of the HBV genome includes four partially overlapping open reading frames (ORFs): *S*, *P*, *C* and *X*. The *S* ORF is composed of *pre-S1*, *pre-S2* and *S* genes and is responsible for the synthesis of three surface proteins included in the HBsAg. The longest *P* ORF encodes the viral enzyme essential for the viral life cycle—HBV polymerase. The whole *C* ORF, including the *pre-C* and *C* regions, is translated into the precursor protein, which yields the hepatitis B e protein (HBeAg), a soluble antigen. The *C* region encodes the hepatitis B core antigen (HBcAg) or C protein, representing the viral capsid’s primary structural protein [7].

Soon after its discovery in the 1960s, it became evident that the newly recognised hepatitis virus was highly variable. Ten serotypes (also known as HBsAg subtypes) were identified based on the amino acid variability of the HBsAg [8]. In the 1980s, HBV genotypes were recognised based on a sequence divergence of more than 8% over the entire genome. Thus far, 10 HBV genotypes (A–J) have been identified [9]. For isolates of genotypes A, B, C, D and E, based on a genome sequence divergence of 4% to 8%, an additional classification of sub-genotypes was introduced, and so far, more than 40 have been recognised [9,10]. The genotypes, sub-genotypes and serotypes have distinct geographical distributions. HBV genotypes and serotypes result from the evolutionary drift of the viral genome as a consequence of a long-term adaptation of the virus to genetic determinants of different host populations. 

On the other hand, HBV is a virus prone to variability that arises spontaneously due to its unique life cycle. The reasons for this spontaneous variability lie in an error-prone viral reverse transcriptase and a very high replication rate. The estimated mutation frequency of HBV is approximately 10-fold higher than for other DNA viruses (1.4 to 3.2 × 10^−5^ substitutions/site/year) [11]. Accordingly, HBV exists as a quasi-species population whose composition is determined by the host immune response and antiviral therapy or vaccination.

HBV also plays a role in natural infection with the hepatitis D virus (HDV). HDV is a satellite virus that does not code envelope glycoproteins but depends on HBsAg to form complete virions and, hence, can infect humans only as a co-infection or a superinfection with HBV. Its genome, a single-stranded RNA, is associated with two isoforms of the hepatitis delta antigen—small (S-HDAg) and large (L-HDAg)—to form ribonucleoprotein (RNP). HDV RNA has >70% internal base pairing, enabling folding into a partially double-stranded rod-like structure. HDV virion assembly depends on the interaction of RNP with HBsAg isoforms [12].

## 3. The Biological Roles of HBsAg Isoforms

Surface proteins are all encoded by a single open reading frame—*S*-ORF divided by three start codons into the *pre-S1*, *pre-S2* and *S* domain [13]. Two sub-genomic mRNAs are responsible for the translation of the surface proteins: 2.4 kb sub-genomic mRNA, transcribed from *pre-S1*, *pre-S2* and *S* domains, for LHB and 2.1 kb sub-genomic mRNA, transcribed from *pre-S2* and *S* domains, for MHB and SHB (Figure 1). Surface proteins’ mRNAs are mainly transcribed from cccDNA but can also be transcribed from the parts of the integrated viral genome.

The three surface proteins share the same C-terminus but have different N-terminal extensions. The SHB consists of 226 amino acids (aa), the MHB protein contains an extra 55 aa at the N-terminus, while the LHB protein has an additional 108 or 119 aa (depending on the genotype) at the N-terminus, relative to MHB. The ratio between LHB, MHB and SHB in the envelope of mature virions is 1:1:4 [14]. The synthesis of envelope proteins is diverse from genome replication and occurs in the endoplasmic reticulum (ER), where they integrate into the ER membrane. Four transmembrane domains are within the S-domain of all three envelope proteins, connected by internal and external loops [15]. In the ER, the three envelope proteins undergo co- and post-translation modification in the form of N-glycosylation. All three proteins are N-glycosylated at asparagine 146 in the S-domain, while MHB is N-glycosylated at asparagine 4 of the pre-S2 domain [16,17]. The LHB shows no N-glycosylation at pre-S2, but its pre-S1 region is myristoylated at the G2 residue [18]. The N-glycosylation sites have significant roles in the viral life cycle. In addition, in HBV genotypes C and D, there is an O-glycosylation site at T37 within pre-S2 [19].

The transmembrane topology of surface proteins is essential for their antigenicity and function in viral morphogenesis and release [20,21]. Unlike MHB and SHB, the LHB protein exhibits two transmembrane topologies. LHB’s pre-S1 and pre-S2 domains can be transported from the luminal to the cytoplasmic side of the ER by using myristoylated residue at position 2 in pre-S1 as an anchor [22]. Approximately half of LHB proteins have their pre-S-domain on the cytoplasmic side of the ER, while the other half have their pre-S on the ER luminal side. During the step of envelope formation, the pre-S1 on the cytoplasmic side is essential for the interaction between the LHB and the newly formed capsid [23]. A short sequence on the junction between pre-S1 and pre-S2 in LHB, called the matrix domain (MD), is crucial for binding with the capsid and forming complete virions [24]. 

The pre-S1 exposed on the luminal side of the ER will end up on the surface of the Dane particle, where aa 2–48 serves as a viral anti-receptor for the bile receptor, sodium taurocholate co-transporting polypeptide (NTCP), recognised as the major HBV attachment molecule on the surface of the hepatocytes [25]. The specificity of binding to NTCP is improved by the myristoylation of the N-terminus of the pre-S1. The HBV virion initially binds to heparan sulphate proteoglycans expressed on the surface of hepatocytes and then with high specificity to NTCP. It was shown that the topology of LHB is changeable in a way that the pre-S1 can switch between the interior and exterior of the virion [26]. Since heparan sulphate proteoglycans can be found on various cell types, it is now believed that LHB does not express pre-S1 on the surface during the initial infection of a naïve host to avoid ineffective attachment before virions can reach the liver.

Unlike LHB, the role of the MHB protein in the viral life cycle remains elusive. MHB is so far known to promote virion secretion but is not essential for virion release or infectivity [27]. Although its role seems dispensable for the viral life cycle, it was pointed out that MHBs may affect the production of virus particles by affecting the ratio of LHBs/SHBs [24]. SHB proteins are the most abundant in the viral envelope. They are required for virion morphogenesis and secretion and contain both B- and T-cell epitopes.

The central core of the S-domain (aa99 to 169) in all three surface proteins is exposed on the surface of virions and is called the major hydrophilic region (MHR). A cluster of B-cell epitopes within MHR (aa 124–147) forms a major antigenic determinant called “a” determinant [8]. The “a” determinant is found in all genotypes and serotypes of HBV and has a highly conserved sequence. This antigenic loop of HBsAg is responsible for a low-affinity interaction between HBV and heparan sulphate proteoglycans (HSPGs) present on hepatocytes at the beginning of the HBV life cycle [28]. Any changes in amino acid sequences within “a” determinant, because of point mutation, deletion or insertion within the *S*-domain, can lead to important changes concerning immunity and protection from HBV infection.

In addition to “a” determinant, the surface proteins contain several other B- and T-cell epitopes crucial for inducing an immune response to HBV infection [20]. Antibodies develop in response to several viral proteins, but only those targeting specific epitopes of the surface proteins are neutralising. Neutralising antibodies target either the antigenic loop (“a” determinant) of all three surface proteins and interfere with the initial attachment to HSPGs or the pre-S region of LHB and block the binding to the specific NTCP host receptor [29]. Accordingly, the presence of anti-HBs antibodies is considered to confer immunity against HBV infection. Interestingly, the pre-S2 region was found to be more immunogenic than the S [30].

The three surface proteins are not only the structural components but can act as immune modulators responsible for establishing chronic infection [31]. It was shown that HBsAg can cause the deregulation of both innate and adaptive immune responses by interacting with immune and non-immune cells. This is, in turn, responsible for the ability of HBV to escape and control the host’s immune system and cause liver damage.

The amount of envelope proteins synthesised in the ER exceeds by far the amount needed for virion assembly. The excess proteins are exported from the cell as non-infectious subviral particles (SVP) [32]. Thus, besides Dane particles, two forms of SVPs, spherical and filamentous, can be found in the blood of infected individuals, where they exceed the number of virions 10,000- to 100,000-fold. The SVPs mainly consist of SHB, while MHB and LHB are present in about 20%. The ratio between LHB, MHB and SHB in filamentous SVPs is the same as in mature virions, 1:1:4, while spherical SVPs contain mostly SHB, less MHB and only traces of LHB [33]. In contrast to the secretory pathway of Dane particles, which requires host factors of the endosomal sorting complex required for transport (ESCRT) and multivesicular bodies (MVBs) for release, spherical SVPs are excreted via the constitutive secretory pathway [34,35]. Unlike spherical SVPs, filamentous SVPs are rich with LHBs, and it is now believed that filamentous SPVs follow the same secretory pathway as Dane particles [36]. The presence of subviral particles does not compromise the entry of complete virions into the hepatocytes, and their assumed role is as a decoy for the immune system, particularly neutralising anti-HBs antibodies [37].

The HBV surface proteins also play a role in natural HDV infection. HDV morphogenesis is based on the interaction between the farnesylated N-terminus of large delta protein (L-HDAg) and SHB [12]. It was shown that the HDV envelope has similarities with spherical SVPs since it has the same HBsAg isoforms as spherical SVPs and probably follows the same secretory pathway [32]. However, further research is needed to elucidate this process since it is known that LHB is required to form an infectious viral particle.

The HBV life cycle functioning depends on a balanced expression of all envelope proteins, and any changes in their production can lead to their intracellular retention, which was shown to have a direct cytopathic effect [38]. In addition, the expression of envelope proteins is essential for regulating cccDNA levels in the nucleus of infected hepatocytes [39]. The rise in the cccDNA pool can also be associated with hepatocyte death [40,41].

## 4. The Impact of HBsAg Isoforms on HBV-Related Liver Phases and Clinical Manifestations

The genetic variability of HBsAg isoforms may play a role in several HBV-related liver phases and clinical manifestations (Figure 2). Most studies about the clinical consequences of HBsAg variants focused on the major hydrophilic region of the *S* gene, especially the “a” determinant. Amino acid changes within the “a” determinant arise from selection or natural variation and can lead to conformational changes, which may lead to consequences such as occult HBV infection and HBV reactivation upon immunosuppression, evasion of anti-HBV immunoglobulin therapy or vaccine-induced immunity [42,43]. They all influence the antigenicity of HBsAg and are sometimes called “immune-escape” mutations. In addition to the mutations in the *S* gene, many *pre-S* mutants have shown a correlation with the clinical implications such as occult infection, liver fibrosis and cirrhosis, fibrosing cholestatic hepatitis (FCH) and HCC development [15,20,21]. Patients with long-lasting HBV infection often have numerous point mutations, insertions and deletions within *pre-S1* and *pre-S2* regions, making them the most variable regions in the HBV genome.

### 4.1. HBsAg Mutants Associated with Immune Evasion

HBsAg mutants capable of immune evasion often result from missense mutations involving only one amino acid residue and seldom from the insertions or deletions of multiple residues. The phenomenon of vaccine escape was first described in the late 1980s in a follow-up study of childhood vaccination. It was revealed that vaccinated children with a strong antibody response to HBsAg could still become HBsAg-positive through HBV infection [44]. The first vaccine-escape mutation associated with this phenomenon was the substitution of a glycine at position 145 by arginine (G145R) [45]. This has become the most widely reported vaccine-escape mutant, but many other substitutions in the “a” determinant have since been associated with evasion of vaccine-induced immunity like T116N, P120S/E, I/T126A/N/I/S, Q129H/R, M133L, K141E, P142S, D144A/E and G145A [46]. HBsAg immune-escape mutants presumably arise in response to immune pressure, such as the presence of anti-HBs antibodies, in resolved infections or after vaccination. However, they can also be found as a minor quasi-species population in non-vaccinated individuals [8]. Thus, it is assumed that they originate from spontaneous replication errors or other driving forces, such as the T-cell immune response. So far, the emergence of these mutations has not had a known negative effect on immunisation programs worldwide. The same escape variants may emerge in patients with graft infections after liver transplantation who receive immunoprophylaxis with monoclonal or polyclonal HBV-specific immunoglobulins (HBIg). In addition, the immune-escape mutations that emerge can be selected during NA therapy. This is due to the overlapping of *P* and *S* genes since *P* gene mutations selected under the pressure of reverse transcriptase inhibitors can be reflected in the *S* gene and change the antigenicity of HBsAg [47].

### 4.2. HBsAg Mutants Associated with Occult HBV Infection

Occult HBV infection (OBI) is defined as the presence of replication-competent HBV DNA (mostly episomal cccDNA) in the liver with or without the presence of HBV DNA in the serum of individuals who test negative for hepatitis B surface antigens (HBsAg) by currently available assays [48]. The molecular basis of OBI lies in the stability and persistence of cccDNA in the hepatocytes. HBsAg may become negative either following the resolution of acute hepatitis B or after decades of HBsAg-positive chronic hepatitis B, spontaneously or induced by antiviral therapy. In all cases, OBI can lead to severe clinical consequences for the infected individual, like the progression of liver fibrosis, the development of HCC or severe infection reactivation upon immunosuppression. In addition, OBI carries the risk of possible viral transmission to uninfected individuals by blood transfusion or organ transplantation [49].

In most cases, OBI arises from the suppression of viral replication and HBsAg expression as a result of either epigenetic mechanisms or the host’s immune control. *PreS/S* gene mutants are responsible for OBI in rare cases by causing the production of modified HBsAg, which cannot be recognised by standard immunoassays [48]. This type of occult infection (often referred to as “false” OBI) is associated with the level of viral replication corresponding to those in overt infection. It carries the highest risk for infection transmission, especially in blood transfusion settings [50,51].

Numerous studies have shown that OBI patients bear a more significant proportion of mutations in the *pre-S/S* region than patients with overt infection. The *S*-region mutations cause a reduction in antigenicity for the detection of HBsAg. Some of the *S* gene mutations associated with OBI are located within and some outside of the “a” determinant: Y100S, Q101R, P105R, T115N, T116N, G119R, P120L, R122P, T123N, C124R/Y, T126I/S, P127H/L, Q129P/R, M133T, Y134C, S136P, C139R, T140I, K141E, S143L, D144A, G145R/A, S167L, R169H, S174N, L175S, V177A and Q181* [52,53,54]. On the other hand, *pre-S1* and *pre-S2* mutations found in OBI alter the production or secretion of surface proteins. The mutations that change the *pre-S2/S* promoter or *pre-S2* start codon cause an imbalance in the production of three surface proteins, which leads to their accumulation in the ER and the impaired secretion of HBsAg [55,56,57].

### 4.3. HBsAg Mutants Associated with Fulminant Hepatitis

In rare cases of acute hepatitis and a significant number of cases of chronic hepatitis in the reactivation phase, HBV infection can manifest with very severe, life-threatening symptoms resulting from massive liver necrosis and multiple organ failure. Fulminant hepatitis (FH) or acute liver failure (ALF) is defined by coagulopathy (international normalised ratio—INR > 1.5) and the presence of hepatic encephalopathy in the absence of chronic, underlying (or prior) disease and a duration of illness of <26 weeks [58]. An acute exacerbation in chronic HBV infection (reactivation), usually upon immunosuppression, may develop in the form of fulminant hepatitis, even though it does not entirely fulfil the definition of FH [59].

Virus-induced liver damage in FH results from an interplay between the virus replication and the host’s defence. Accumulating evidence in recent years associates viral genotypes and mutations in different genome regions with the onset of FH [60,61]. All viral factors contributing to the enhanced replication or induction of a more potent immune attack could be responsible for the development of FH.

Since MHB is not crucial for viral survival, MHB-deficient mutants are readily selected. The *pre-S2* region can accumulate mutations and even deletions that can be tolerated because this region overlaps the spacer domain of the *P* gene, which can be changed without affecting the polymerase activity [62]. Following this, studies have reported that HBV isolates from patients with FH displayed a double nucleotide mutation in the start codon (ATG to ACA) of the *pre-S2* region that prevented the synthesis of the MHB [63,64,65]. The cessation of MHB production can favour the development of FH in several ways [66]. First, it causes an imbalance between three surface proteins because LHB is overexpressed compared to MHB and SHB. The retention of HBV surface proteins was associated with FH in a transgenic mouse model, where these accumulated proteins led to the extreme sensitivity of hepatocytes to interferon-gamma produced by the cytotoxic T lymphocytes [38]. Abundant viral antigens in the hepatocytes led to the hypothesis of a direct cytopathic effect of the virus responsible for liver injury in FH, FCH and cirrhosis cases [67,68,69,70]. The specific T- and B-lymphocyte response to the MHB protein is an important early event in the immune response to HBV infection, which leads to the conclusion that the absence of this protein may result in the immune system’s inefficient neutralisation of the virus [20,71]. Finally, the cccDNA pool is regulated by the expression of surface proteins so that when the production of surface protein lessens, the amplification of cccDNA increases, eventually causing hepatocyte death [40,41,55].

Within the *S*-region, immune-escape mutations within MHR and deletions outside MHR have been associated with the occurrence of FH [71,72]. A premature stop codon at position 216 (L216*) of the *S*-domain causes the production of truncated surface proteins that accumulate in the ER and induce apoptosis, contributing to FH.

### 4.4. HBsAg Mutants Associated with Liver Fibrosis, Cirrhosis and FCH

As previously mentioned, HBsAg mutations can cause an imbalance in the production and secretion of HBsAg isoforms and their intracellular accumulation. This may prove to be directly cytotoxic to hepatocytes and represent an additional mechanism of liver injury, which can lead to the progression of fibrosis and the development of cirrhosis [69,70]. The mutations associated with these conditions are located within the pre-S regions, thus affecting LHB and MHB isoforms. It was suggested that the deletion within the pre-S2 region could be used as a predictor of fibrosis progression [70]. The proposed direct cytopathic effect caused by the accumulation of mutated LHB and MHB isoforms was also associated with the rapidly progressive form of liver injury-fibrosing cholestatic hepatitis (FCH), which often develops in transplanted, immunocompromised patients [67,68].

### 4.5. HBsAg Mutants Associated with HCC

Primary liver cancer (hepatocellular carcinoma) represents a significant public health problem, being the second most frequent cause of cancer-related mortality worldwide [73]. HBV is the most common cause of HCC and can contribute to its development by two mechanisms. HBV DNA can integrate into the host genome and act as an insertional mutagen, causing the activation of proto-oncogenes, or different HBV proteins can interact with numerous host proteins and modulate several cellular signalling pathways related to HCC [74].

The association between mutations of HBV surface proteins and carcinogenesis has been well established [20,75,76,77]. HCC was commonly related to HBV isolates with deletions of the C-terminal part of the *pre-S1* region and/or deletions of the N-terminal domain of the *pre-S2* region and/or mutations in the *pre-S2* start codon [76]. As in the previously described mechanism associated with fulminant hepatitis, these mutations can lead to an imbalance in the production of three surface proteins and their retention within the hepatocytes. All three surface proteins, particularly *pre-S1* and *pre-S2*-deletion mutants, can be retained within the ER, which induces ER stress signalling pathways [78,79]. Histologically, hepatocytes with accumulated surface proteins within the ER are recognised as ground-glass hepatocytes, which are characteristic of chronic HBV infection but are also predecessors of HCC [80,81]. The ER stress initiates three oncogenic signalling pathways [20,82]. The first one involves oxidative stress to induce DNA damage and genomic instability. The second one includes an increase in vascular endothelial growth factor-A (VEGF-A) expression, which engages VEGF receptor 2 (VEGFR-2) to initiate the protein kinase B/mammalian target of rapamycin (Akt/mTOR) signalling and results in the promotion of cell proliferation [83]. The third pathway upregulates cyclooxygenase-2 (COX-2) through the nuclear factor kappa-light-chain-enhancer of activated B cells (NF-κB) and p38 mitogen-activated protein kinase (MAPK) signalling to enhance anchorage-independent cell growth [84]. *Pre-S2* deleted proteins can additionally initiate the overexpression of cyclin A through the calcium-dependent protease μ-calpain A to induce centrosome overduplication and chromosome instability [85]. Independent from ER stress, *pre-S2* mutants display proto-oncogenic potential in direct binding to Jun activation domain-binding protein 1 (JAB1), thus activating the degradation of cyclin-dependent kinase (Cdk) inhibitor p27, Rb hyperphosphorylation and cell cycle progression [86]. Pre-S2 deleted proteins can also inhibit DNA repair through importin α1/NBS1 and enhance cell survival through the enhanced expression of the apoptosis regulator B cell lymphoma-2 (Bcl-2) [87].

C-terminally truncated MHB proteins (MHBst) retained in the ER show transforming properties. They can promote a protein kinase C (PKC)-dependent activation of the c-Raf-1/MEK/Erk2 signalling cascade, thereby promoting hepatocyte proliferation [88]. MHBst exhibits transactivation properties due to its specific topology. The pre-S domain of MHst is directed toward the cytoplasm, unlike the structural MHB, where it can trigger intracellular signal transduction cascades. It can upregulate the human telomerase reverse transcriptase (hTERT) promoter, thus enhancing telomerase activity and advancing the development of HCC [89].

HCC has also been associated with S-domain mutations outside the “a” determinant. The reverse transcriptase mutation A181T, selected during long-term NA therapy, corresponds to a stop codon mutation in the overlapping surface proteins at position 172 (W172*). The resulting truncation of surface proteins was shown to have a secretory defect and can transactivate oncogene promoters [90]. Despite the many favourable effects of NA therapy on HBV-induced liver disease progression, it does not entirely reduce the risk of HCC development because suppression of viral replication does not affect S gene transcription and protein synthesis [3,20].

## 5. Diagnostic Applications of HBsAg Isoforms

HBsAg is the oldest antigen used in diagnosing HBV infection, and it originates from both HBV mini-chromosomes (cccDNA) and randomly integrated HBV DNA into the host genome. It is found in complete virions, subviral particles, and incomplete virions (containing pre-genomic RNA or double-stranded linear DNA). It is a target for the routine diagnosis of both acute and chronic infection as the qualitative or quantitative detection of total HBsAg. In recent years, additional methods have been under development, like the quantitation of HBsAg isoforms and the quantitation of O-glycosylated MHB (Figure 2).

### 5.1. Qualitative and Quantitative HBsAg Detection

The qualitative detection of serum HBsAg remains the principal biomarker for screening and establishing the initial diagnosis of HBV infection. In addition, a spontaneous or treatment-induced loss of HBsAg is considered the crucial endpoint, called the functional cure. Thus, in the past decade, HBsAg quantitation (qHBsAg) has been introduced in clinical practice to predict the possibility of a functional cure. This method’s other widely used applications are the follow-up of treatment response and the evaluation of liver disease progression, particularly the potential of HCC development [91].

Enzyme quantitative immunoassays usually perform HBsAg quantitation with an analytical sensitivity of 0.05 IU/mL. New ultrasensitive methods with sensitivities of 0.005 IU/mL have been developed recently. Their application seems justified in settings like OBI for diagnosing and assessing reactivation risk [92,93,94].

HBsAg levels differ between phases in the natural course of chronic HBV infection by being higher in HBeAg-positive than HBeAg-negative phases. The highest levels (>4 log IU/mL) are recorded in HBeAg-positive infection, and the lowest (~2 log IU/mL) in HBeAg-negative infection [95]. During HBeAg-positive phases, they reflect cccDNA activity and correlate well with HBV DNA. On the other hand, this correlation is weaker in HBeAg-negative phases since the primary source of HBsAg is subviral particles derived from integrated HBV DNA [96]. The qHBsAg also depends on the viral genotype and the presence of pre-S mutations, with higher levels in genotypes A and E and lower levels if the pre-S mutants dominate [6,95]. Low baseline levels of HBsAg and anti-HBc and substantial HBsAg decline during PEG-IFN and NA therapy are predictors of HBsAg seroclearance [97,98]. During NA therapy, HBV DNA levels decrease rapidly, while qHBsAg has a slower decline, and in HBeAg-negative phases, the correlation with HBV DNA is very low. The level of HBV RNA is another surrogate marker of cccDNA transcriptional activity, which shows similar kinetics to qHBsAg, with the highest levels in HBeAg-positive infection and lowest in HBeAg-negative infection. As in qHBsAg, it has higher values in HBeAg-positive than HBeAg-negative phases and declines much slower than HBV DNA during NA therapy [95,97]. It is still present in virological suppression, where HBV DNA is suppressed, but the HBsAg is still detectable. Since HBV RNA is measurable long after the suppression of HBV DNA, it can be used to predict HBsAg loss [6,95].

### 5.2. The Quantitation of HBsAg Isoforms

The proportion of HBsAg isoforms in different stages of HBV infection has been a research subject for decades. It was shown that this proportion was not the same in different donors and was possibly dependent on the type or phase of infection [99,100,101]. In recent years, the presence and balance of different HBsAg isotypes have begun to be investigated to gain more biomarkers to determine phases of chronic infection, predict treatment endpoints, and evaluate liver disease progression.

First, some studies demonstrated how the morphology and composition of secreted subviral particles were viral genotype-dependent [102,103]. The relative amounts of LHBs and MHBs compared to SHBs were found to be higher in genotypes B and D and lower in A, C and E [102]. While studies in vitro recorded more elevated amounts of LHB than MHB in genotypes A-D and J [104,105], the study involving human subjects observed higher amounts of MHB than LHB in genotypes C, D and E [102]. Further, the disease stage, HBeAg-positive or -negative, was proposed to influence genotype-dependent HBsAg composition. In studies including HBeAg-positive patients, one reported higher absolute levels of MHB and LHB in genotype B than C [106], and another showed that genotype B-infected patients had higher levels of LHB than patients infected with genotypes A and D [107]. In the most recent study with the HBeAg-negative patient cohort that included genotypes A, D and E, genotype E was associated with a higher level of total HBsAg, and all its components and genotype D displayed a connection with the highest levels of LHB and MHB [108]. Also, in HBeAg-negative patients, HBsAg-containing particles showed higher density in genotypes B and D [102]. However, the composition and size of subviral particles were not associated with genetic changes in the *pre-S1* and *pre-S2* regions. In conclusion, the proposed genotype-specific viral immunogenicity and infection pathogenesis may be attributed to the genotype-specific proportion of HBsAg isoforms.

The distinction between phases of chronic HBV infection is crucial in clinical practice, particularly the differentiation of inactive “infection” and active “hepatitis” stages that bring the risk of liver disease progression or reactivation. There are indications that LHB and MHB protein levels may be more helpful for this purpose than total HBsAg. The decline in LHB rations was shown to be a stronger predictor of the “inactive carrier” stage than the decline of total HBsAg level [103]. Since LHB is more present in virions and filamentous SVPs than spherical SVPs, the decrease in the ratio of virions to spherical SVPs in low viraemic phases may be responsible for the observed LHB decline. Also, all immune response components act more effectively against pre-S1-containing particles in these infection stages. Another study confirmed the value of LHB quantitation as a complementary marker to determine different stages of infection [109]. The LHB level was found to be much higher in hepatitis than in infection phases or healthy blood donors. The correlation between LHB and HBV DNA levels varied depending on HBsAg level. It was satisfactory in very low (<0.05 IU/mL) and very high (>1000 IU/mL) HBsAg levels, where the negative predictive value of LHB for HBV DNA was 100%. This correlation is attributed to LHB translation, independent of MHB and SHB. LHB has its own mRNA, which is not transcribed from integrated DNA but only from cccDNA.

A higher ratio of MHB to other surface proteins was found in the early stages of acute infection [19,103,110]. Although the exact role of MHB in the HBV life cycle is still unclear, it has been speculated that it has an immunomodulatory role like HBeAg. This role would be necessary in early infection and become unnecessary later in chronic infection when a robust immune response is established. MHB has also been indicated as a player in the process of carcinogenesis [88,89]. C-terminally truncated MHB can act as a trans-activator and upregulate promoter of human telomerase reverse transcriptase (hTERT), thus enhancing telomerase activity and advancing the development of HCC [89]. The pre-S2 protein was also found to act as a trans-activator of forkhead box protein 3 (FOXP3), whose increased expression was identified in HCC cells [111]. The study involving patients with suppressed viral replication showed a >25% increase in MHB in patients who developed HCC compared to those without HCC [112]. DNA integration is the critical step in HBV-mediated carcinogenesis, and the relative ratio of MHB could be increased because it could be produced from integrated HBV DNA lacking the start codon for LHB. In fact, the structural arrangement of the integrated HBV DNA form does not necessarily affect the expression of MHB and SHB, but the expression of LHB may be absent because parts of *pre-S1* may be lost during integration. The rise of MHB might also result from a mutation in the *pre-S2* promoter that upregulates the expression of this protein. Since the prediction of HCC risk under anti-HBV treatment is still challenging, MHB might represent a new biomarker. In contrast, another study reported an association between higher pre-S1 levels and liver inflammation and identified a higher pre-S1/HBsAg ratio as a predictor of HCC development [113]. Finally, in the study that included patients with developed HCC who underwent hepatectomy, the authors reported lower levels of LHB and total HBsAg in HCC than in chronic infection. They identified that the elevated serum level of LHB was significantly associated with worse disease-free and overall survival rates in patients with HCC [114].

The ability to predict therapy endpoints was extensively researched for all currently known biomarkers, and now it is beginning to be investigated for the quantitation of HBsAg isoforms. In addition to total HBsAg quantitation, as the marker primarily recommended for functional cure prediction, the HBsAg isoforms were evaluated as potential new biomarkers of HBsAg loss during therapy. A significant association was found between HBsAg composition before and during NA or PEG-IFN treatment and subsequent HBsAg loss [107]. Patients who achieved HBsAg loss had a progressive decrease in LHB and MHB ratio early during therapy with both NA and PEG-IFN, and the proportions of LHB and MHB were more reliable markers of HBsAg loss compared to the level of HBsAg or HBV DNA. The ratio of LHB and MHB decreased rapidly in HBeAg-positive patients who achieved functional cure but was not changed in patients who only reached HBeAg seroconversion or had no serological response. In NA-treated patients, the decrease in the MHB proportion was a stronger predictor than that of LHB. On the other hand, Rinker et al. reported that all three surface proteins were significantly lower in PEG-IFN responders than non-responders and did not show any prediction advantage compared to the level of total HBsAg [106]. A similar conclusion was reached by Rodgers et al., who investigated three cohorts of HBV-infected patients: those in acute infection, those who were treatment-naïve and patients on NA therapy [110]. They found a constant agreement of results using qHBsAg and levels of total HBsAg, LHB and MHB. Total HBsAg, LHB and MHB levels were significantly higher in HBeAg-positive than in HBeAg-negative phases of infection, but their proportions were similar. The changes in LHB and MHB levels were proportional to levels of total HBsAg in patients on NA therapy regardless of response and HBeAg status.

Finally, some studies attempted to investigate the correlation between levels of HBsAg isoforms and the severity and outcome of chronic HDV infection [103,108]. A higher proportion of LHB was observed in HDV-infected patients of both studies, and patients with higher HDV viremia also had higher levels of qHBsAg and LHB than patients with undetectable HDV RNA. Higher LHB levels were also recorded in HDV-infected patients with developed cirrhosis. The composition of HBsAg was similar regardless of NA therapy or HBeAg status. The importance of LHB in active chronic HDV infection can be explained by its pivotal role in both viruses’ entry into hepatocytes. On the other hand, despite similar qHBsAg levels, patients with unfavourable outcomes like HCC, liver decompensation and mortality had higher levels of MHB at baseline and during the follow-up. In line with this, patients with lower MHB levels at baseline were likely to achieve HDV RNA undetectability. The association of MHB with unfavourable outcomes can also be attributed to MHB’s immunomodulatory role, mainly investigated in chronic HDV infection.

Despite interesting conclusions and promising usefulness, the clinical application of HBsAg composition determination is limited due to the lack of a standardised assay for their quantitation. It is often performed by an in-house quantitative sandwich ELISA or Western blot, using different isoform-specific antibodies. Thus, occasional confronting results can be attributed to this need for reproducibility and correlation between different studies and the lack of automatised methods.

### 5.3. Following Kinetics of Serum O-Glycosylated MHB

Available HBsAg assays cannot distinguish between envelope proteins found on the surface of complete, DNA-containing virions and the excess of envelope proteins organised as SVPs. The partial O-glycosylation of the preS2-domain of MHB was identified as a characteristic of genotypes C and D [19]. Recently, this characteristic was identified as a primary component of mature DNA-containing virions in contrast to MHBs present on SVPs, thus making it a potential biomarker of viral kinetics during the natural history and therapy of HBV infection [115]. First, an in-house immunoassay using the specific antibodies against O-glycosylated pre-S2 in MHB was developed, and now there is a commercial ELISA (RCMG Inc., Tsukuba, Ibaraki, Japan) for quantitation of O-glycosylated MHB (HBsAgGi). It was reported that O-glycosylated MHB serum levels decreased significantly after 48 weeks of NA therapy, although to a lesser extent than HBV DNA [115,116]. More importantly, O-glycosylated MHB serum levels remained detectable even in patients who achieved HBV DNA undetectability after 48 weeks of therapy. Murata et al. then further examined the O-glycosylated HBsAg-binding fraction extracted by immunoprecipitation from HBV DNA-negative patients and found still quantifiable HBV DNA and RNA [115]. As this primarily reflected the number of RNA-containing virions in viral-suppressed patients, it was suggested that this immunoassay could act as a surrogate marker for viral RNA and, thus, cccDNA activity. This is further supported by the loss of correlation of HBsAgGi with HBV DNA and HBsAg after 48 weeks of NA therapy and the much slower decline of this marker in this setting. In addition, Okumura et al. [116] demonstrated that an increase in HBsAgGi/HBsAg ratio was associated with HCC history, following the previously mentioned association of an increased MHB level with HCC [112]. However, the major limitation of the HBsAgGi marker is its possible utility only in infection with two HBV genotypes.

## 6. The Applications of HBsAg Isoforms in Prevention and Therapy

### 6.1. Generations of Anti-HBV Vaccines Based on HBsAg Isoforms

Since the discovery of HBV, HBsAg has been the primary target for vaccine development due to its immunogenic properties (Figure 2). This was based on the observation that the presence of anti-HBs antibodies could provide protection against the infection [117]. The first vaccine, based on a heat-treated form of HBV, was developed in 1969 [118]. The idea of a safer subunit vaccine, composed only of HBsAg, was soon realised by creating the first widely used plasma-derived vaccine in 1981 [119]. It was based on the purification of HBsAg directly from the blood of asymptomatic HBV carriers. The composition of the plasma-derived HBsAg subviral particles varied depending on the purification and inactivation procedures. They mainly comprised SHB, but some contained a small quantity of MHB [120,121]. Several million individuals were vaccinated with these first-generation vaccines, which were reported to be safe and provide excellent protection rates [118,122].

Safety concerns about the use of human plasma-derived products grew in the era of ongoing HIV pandemics, and because of the advances in recombinant DNA technology, the second generation of recombinant HBV vaccines was developed in the mid-1980s. They were produced by expressing HBsAg in genetically engineered yeast cell lines transfected with the HBV S gene [123,124]. After the extraction of the expressed proteins from yeast cells, HBsAg self-assembles into 22 nm spherical subviral particles, almost identical to those found in the serum of infected individuals [125,126]. Yeast-derived SVPs contain almost exclusively SHB and are not N-glycosylated at the N146 position within the S-domain, unlike SVPs produced by mammalian cell lines or found in human blood [127,128]. No significant difference was observed between the anti-HBs response elicited by yeast- and plasma-derived vaccines. However, the efficient production and improved safety of second-generation vaccines made them the most widely used anti-HBV vaccine for almost four decades [129].

Despite second-generation vaccines’ overall seroprotection of 90–95%, there are still 5–10% of non-responders, and their number varies significantly in special population groups [130,131,132]. A new vaccine type was developed to improve immune response against HBV in groups of non-responders. It was approved in 2017 and contained yeast-derived SVPs with Toll-like receptor 9 (TLR 9) agonist cytidine phospho-guanosine oligonucleotide (CpG 1018) as an adjuvant [133,134]. It was a two-dose regime vaccine, able to induce an innate immune response and the production of cytokines such as interleukin-12 and interferon-alpha. It proved to be a useful additional option for protecting non- or hypo-responders due to its shorter schedule, earlier seroprotection and higher seroprotection rates.

The third-generation vaccines were also designed aiming to improve immune response in special populations. The idea was to produce a vaccine that included more HBsAg isoforms to introduce pre-S1 and pre-S2 sequences because of their essential functions in the viral life cycle and roles in protection stimulation. A pre-S1 protein sequence can induce the production of anti-pre-S1 antibodies, which can hinder attachment to the NTCP receptor, the cellular receptor for both HBV and HDV [135,136]. Also, pre-S1 and pre-S2 sequences can provide additional B and T-cell epitopes and help overcome non-responsiveness to the S sequence alone [137].

The first third-generation vaccines were created in the 1990s, again by recombinant DNA technology in HBV-transfected mammalian cell lines (hamster ovary cells –CHO and mouse-derived cells) [138,139]. After the initial introduction, they were not widely used due to high manufacturing costs. Reports indicated that vaccines containing SHB and MHB or all three surface proteins were superior to mono-antigenic vaccines containing only the SHB protein, particularly in specific groups like individuals >45 years or older [140,141,142,143,144]. In 2021, a new three-antigen vaccine (Sci-B-Vac) was approved in the US, EU and other countries. It comprises all three surface antigens and their glycosylated isoforms, and this was shown to stimulate a faster and longer-lasting immune response [145]. Glycosylation was shown to significantly influence MHC presentation through the stimulation of protein antigen uptake and proteolytic processing. In addition, glycans are readily recognised by C-type lectin receptors on antigen-presenting cells.

### 6.2. New Antiviral Strategies Targeting HBsAg Isoforms

The successful inhibition of HBsAg production from cccDNA and integrated viral DNA is needed to achieve HBsAg seroclearance as the primary endpoint of therapy for HBV infection. There are three new approaches targeting HBsAg to attain this goal: blocking HBsAg biological functions like viral entry and HBsAg release and blocking HBsAg production by interfering with viral RNA (Figure 2) [32,146].

The viral entry inhibitors act by blocking the NTCP receptor on hepatocytes. The first entry-inhibitor, bulevirtide (Myrcludex B, Hepcludex), is a synthetic N-acylated pre-S1 lipopeptide that can attach to NTCP, blocking the virus’s entry mechanism. It inhibited HBV infection in in vitro and in vivo experiments [147,148]. Since HDV uses the same entry receptor, bulevirtide was subsequently approved for the treatment of compensated chronic HDV infection, while it is still under investigation for HBV mono-infection [149]. However, blocking the receptor with other biological functions could be associated with side effects that still have to be thoroughly evaluated [150].

HBsAg-release inhibitors exhibit their action by inhibiting viral assembly and release from hepatocytes. Nucleic acid polymers (NAPs) have a unique ability to block the secretion of HBsAg from HBV-infected hepatocytes [32]. They selectively target the assembly and secretion of spherical SVPs without affecting the same processes of HBeAg or Dane particles. In the clinical setting, NAP therapy resulted in rapid declines of HBsAg, HBsAg seroconversion, and declines in HBV DNA and RNA [151]. The results were even more promising when NAPs were combined with tenofovir disoproxil fumarate (TDF) and PEG-IFN [152]. The NAPs combination therapy was also shown to improve levels of HDV RNA in HBV/HDV co-infected patients [153]. HBsAg isoform changes during NAPs therapy in HBV/HDV co-infected patients displayed a more rapid clearance of SHB than other HBsAg isoforms in patients with strong total HBsAg level declines [154]. This selective clearance of subviral particles was consistent with previous NAP studies. The trace of total HBsAg rebound in some patients consisted primarily of SHB and MHB and might reflect only HBsAg isoforms derived from integrated DNA.

Antiviral drugs that interfere with viral RNA are designed to block antigen production. Small interfering RNAs (siRNA) bind and eliminate complementary mRNA in the cytoplasm of infected hepatocytes, and antisense oligonucleotides (ASO) initiate the cleavage of all HBV RNAs in the nucleus and cytoplasm [32]. These new compounds were associated with HBsAg responses but need to be further investigated and may play a role as part of the combination therapies for achieving a functional cure.

### 6.3. Monoclonal Antibodies Targeting HBsAg Isoforms

The first neutralising antibodies against HBV infection were isolated some 40 years ago, but only recently, monoclonal antibodies against HBsAg and its components began to be investigated as potential antiviral molecules that can contribute to achieving a functional cure [155]. Only a handful of monoclonal anti-HBs antibodies have been clinically evaluated so far. One promising result is VIR-3434, which targets a conformational epitope within the antigenic loop of all three surface proteins. Its mechanisms of action include the entry inhibition of all 10 HBV genotypes, delivering HBsAg to antigen-presenting cells that could renovate adaptive T and B cell responses via a vaccinal effect and the reduction of circulating HBsAg [156].

### 6.4. HBsAg Isoforms as Vectors for Other Therapeutic Molecules

Recombinant HBsAg isoforms can be vectors for delivering other therapeutic molecules like drugs, genes or proteins to hepatocytes and HCC cells [21,157]. Pre-S1, with its receptor-binding properties, and pre-S2 domain, with roles in penetration and uncoating, are the most successful candidates for hepatocyte delivery systems. The delivery system can improve the therapeutic effect of specific molecules and reduce side effects on tissues other than the liver. In research, SVPs have been exploited as carrier platforms for antigenic sequences of different viruses (HIV-1, dengue, HCV, poliovirus, influenza A), bacteria (*Helicobacter pylori*) and protozoa (*Plasmodium*) [128].

## 7. Conclusions and Future Directions

HBsAg isoforms are involved in several biological functions during HBV infection and have a vital impact on outcomes of HBV infection. Their role in viral attachment to hepatocytes and establishment of persistent infection is well documented. A compelling genetic variability of *pre-S/S* domains was often associated with the pathogenesis of different HBV-related liver phases and clinical manifestations. More conclusive evidence can be gathered by future research on the relationship between the presence and presentation of different surface antigen components and specific clinical entities such as occult hepatitis and HBV reactivation, fulminant hepatitis and HCC development.

In the present-day management of chronic HBV infection, there is a constant need for new non-invasive biomarkers, which would reflect the virus’s intrahepatic activity and predict the achievement of therapy endpoints. Initial research has already shown promising results in utilising HBsAg isoforms instead of quantitative HBsAg for correctly evaluating chronic infection phases and predicting a functional cure. The ratio between surface components was shown to be indicative of specific outcomes of HBV and HDV infections, and the potential use of HBsAg isoform quantitation as a diagnostic marker is likely to be expected. Thus, developing standardised and universally available quantitation assays is needed.

Since current therapeutic protocols are unable to achieve a sterilising cure in chronically infected patients and the rate of functional cure is still not satisfactory, new therapeutic strategies are directed towards the suppression of viral antigens and immune modulation. HBsAg isoforms will be important targets for developing new therapies because they are crucial in eliciting all types of immune responses during infection. All HBV surface proteins were included in different vaccines developed over time and, because of their immunogenic properties, will probably be part of future vaccine strategies as well.

## Figures and Tables

**Figure 1 pathogens-13-00046-f001:**
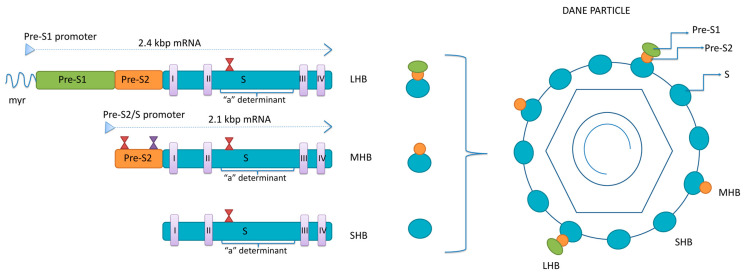
The structure of HBsAg isoforms. LHB—large surface protein; MHB—middle surface protein; SHB—small surface protein; myr—myristoylated position 2 within pre-S1 region; I-IV—transmembrane domains within S region of all three proteins; red hour-glasses—N-glycosylation sites at position 146 within the S region of all three proteins and four within pre-S2 of MHB; violet hour-glass—O-glycosylation site at position 37 within pre-S2 of MHB of genotypes C and D.

**Figure 2 pathogens-13-00046-f002:**
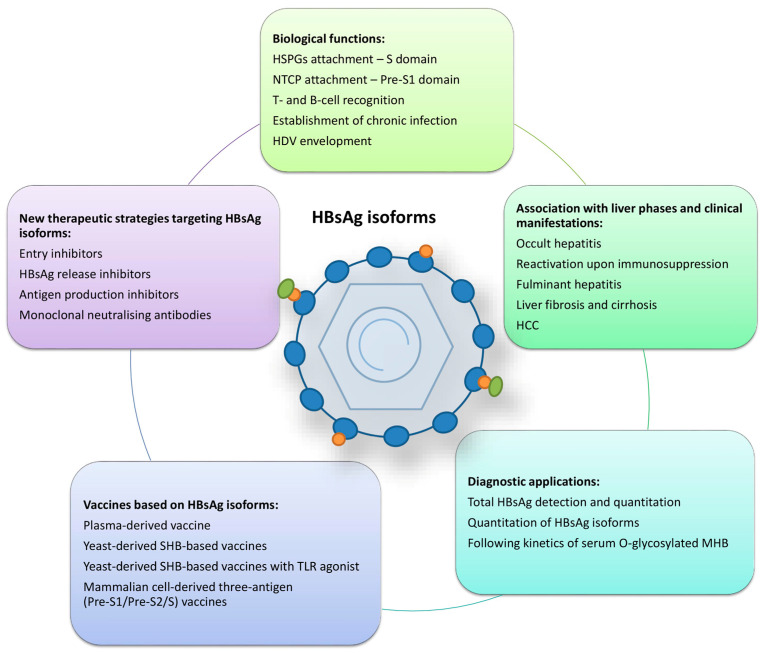
Summarised roles of HBsAg isoforms. HSPGs—heparan sulphate proteoglycans; NTCP—sodium taurocholate co-transporting polypeptide; HDV—hepatitis D virus; HCC—hepatocellular carcinoma; MHB—middle surface protein; TLR—toll-like receptor.

## Data Availability

Not applicable.

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
