# Peer review of "Hepatitis B Surface Antigen Isoforms: Their Clinical Implications, Utilisation in Diagnosis, Prevention and New Antiviral Strategies"

_pathogens, 2024, doi:10.3390/pathogens13010046_

Round 1
Reviewer 1 Report
Comments and Suggestions for Authors
It is a comperhensive review including detailed focused point related to HBs Ag and its role in diagnosis, therapy and vaccine. The review includes the following points: a) HBV molecular virology and genetic variability. b) Biology of HBs Ag
c) Link between HBs Ag and diseases such fluminant hepatitis, heptaocellular carcinoma,..etc
d) HBs Ag as a diagnostic markers. e) Hbs Ag is therapy and vaccine.
The following points should be addressed to improve the quality of the review
1) In section 4.1: HBsAg mutants associated with immune evasion. The author should provide the causes and resons for emergence of vaccine escape mutants such as vaccine program, causes of mutations, other causes.
2) In section 4: please provide HBV in acute HBV infection. The author provided fluminant hepatitis, not acute HBV infection that not progress to FHF. Also in chronic HBV and in cirrhosis and fibrosis
3) in section 5: the link between HBs Ag and other HBV markers used in diagnosis should be provided such as HBe Ag, and anti-HBc Ab
4) in Line 385-390. The correlation between HBs Ag and HBV DNA or RNA should be in details.
What is the link between HBs Ag and RNA????
5) Manuscript need language editing
Comments on the Quality of English Language
Moderate language editing
Author Response
Response to Reviewer 1
Comments and Suggestions for Authors
It is a comperhensive review including detailed focused point related to HBs Ag and its role in diagnosis, therapy and vaccine. The review includes the following points: a) HBV molecular virology and genetic variability. b) Biology of HBs Ag
- c) Link between HBs Ag and diseases such fluminant hepatitis, heptaocellular carcinoma,..etc
- d) HBs Ag as a diagnostic markers. e) Hbs Ag is therapy and vaccine.
The following points should be addressed to improve the quality of the review
1) In section 4.1: HBsAg mutants associated with immune evasion. The author should provide the causes and resons for emergence of vaccine escape mutants such as vaccine program, causes of mutations, other causes.
We thank the reviewer for the valuable comments and suggestions. The suggested information was added.
2) In section 4: please provide HBV in acute HBV infection. The author provided fluminant hepatitis, not acute HBV infection that not progress to FHF. Also in chronic HBV and in cirrhosis and fibrosis
We added information about HBsAg mutations associated with cirrhosis and fibrosis, as suggested. The acute, uncomplicated infection was not included since, although the discovered mutations during this phase are abundant, we could not find evidence of their clinical implications other than the already discussed immune- and vaccine-escape phenomena and fulminant hepatitis.
3) in section 5: the link between HBs Ag and other HBV markers used in diagnosis should be provided such as HBe Ag, and anti-HBc Ab
The information about the correlation between HBsAg and other biomarkers was added.
4) in Line 385-390. The correlation between HBs Ag and HBV DNA or RNA should be in details.
What is the link between HBs Ag and RNA????
The correlation between HBsAg, HBV DNA and HBV RNA was explained in more detail.
5) Manuscript need language editing
The language was edited with the help of a native-speaker scientist.
Reviewer 2 Report
Comments and Suggestions for Authors
Major comments
This is an interesting review, especially the parts referring to virological data. However, since it refers to HBsAg proteins, its needs to be carefully revised and focus on the scarce literature on HBsAg isoforms, not HBsAg in general.
Throughout the manuscript, including the title, I suggest to use the term HBsAg proteins or HBsAg isoforms since these are the most commonly used denominations, and avoid the term HBsAg components. I also suggest to avoid provide information on HBV mutants and just focus on HBsAg proteins.
Minor comments
Abstract
- “several HBV-related liver diseases” I suggest to substitute diseases by phases.
- I suggest to ease up some affirmations about HBsAg isoforms since many of their functions have not been confirmed so far. For instances, “ Genetic variability of HBsAg components plays a considerable role” I suggest to add a “may”.
- “ for developing generations of HBV”- Omit generations.
- “ Initial research has already shown promising results in utilising HBsAg components instead of a whole HBsAg” Substitute whole HBsAg by quantitative HBsAg
- “ This review aimed to summarise numerous biological, diagnostic, preventive and therapeutic roles of HBsAg components and emphasise aspects deserving further research.” I suggest modifying this sentences by “this review aimed to summarise the current evidence on the potential usefulness of HBsAg proteins, the limitations for their quantification and aspects deserving further research”
Introduction
- “The HBV infection evolves into acute or chronic” Avoid this sentence.
- I suggest to avoid the previous terms for the HBV phases (i.e. “(“immune tolerant” phase)” or, at least, add before “previously named as”.
- “HBsAg-negative phase (“occult HBV infection”). These two terms are not the same; delete the term “occult HBV infection”.
- I suggest modifying the last paragraph of this section since it is a copy of the abstract.
HBV molecular virology
- “adw, adr, ayw, and 84 ayr. The major antigenic determinant called “a” was unique in all serotypes, while later 85 discovered subdeterminants extended the number of subtypes to 10: ayw1, ayw2, ayw3, 86 ayw4, ayr, adw2, adw3, adwq, adr, and adrq−” This part should be reduced.
- I would suggest to create a new section with information regarding HDV.
Impact of HBsAg components on HBV-related diseases
This section refers to HBV mutants, not HBsAg proteins, so I suggest to delete it and to center the manuscript in data on HBsAg proteins.
Quantitation of HBsAg isoforms
Add information about the techniques for its performance and the lack of automatized methods.
6.1. Generations of anti-HBV vaccines based on HBsAg components
I suggest to reduce this section.
6.2. New antiviral strategies targeting HBsAg components
This section can also be reduced since it refers to HBsAg, not its proteins.
Future directions
“More conclusive evidence can be gathered by future research on the relationship between the variability of surface antigen components and specific clinical entities such as occult hepatitis and HBV reactivation, fulminant hepatitis and HCC development.” Do not mix the potential role of HBsAg proteins and HBV mutants.
Comments on the Quality of English Language
None
Author Response
Response to Reviewer 2
Major comments
This is an interesting review, especially the parts referring to virological data. However, since it refers to HBsAg proteins, its needs to be carefully revised and focus on the scarce literature on HBsAg isoforms, not HBsAg in general.
Throughout the manuscript, including the title, I suggest to use the term HBsAg proteins or HBsAg isoforms since these are the most commonly used denominations, and avoid the term HBsAg components. I also suggest to avoid provide information on HBV mutants and just focus on HBsAg proteins.
We thank the reviewer for the valuable comments and suggestions. According to the suggestion, we replaced the term “HBsAg components” with HBsAg isoforms” in the title, Figure 2 and throughout the text. However, we do believe that the HBsAg mutations should be addressed in this review for the following reasons:
Since HBV is a variable virus, the mutations are a part of its arsenal for causing specific clinical conditions. Mutations in different domains of the S gene can affect the synthesis and size of different HBsAg proteins and change their biological roles, thus contributing to the development of specific clinical manifestations. The presence of mutations can result in the synthesis of truncated forms of one of the surface proteins, which was associated with the pathogenesis of occult hepatitis and HCC. In some cases, the mutations can cause the abolishment of synthesis of one surface protein, as was suggested to be part of fulminant hepatitis development. The imbalance in surface protein production is responsible for the accumulation of one or more surface proteins in the ER, which is responsible for ER stress and possible direct cytotoxic effect associated with several clinical conditions, such as fulminant hepatitis, progression of fibrosis, development of cirrhosis, HCC and fibrosing cholestatic hepatitis.
In addition, we did try to reduce the unnecessary parts in chapters about HBV vaccines based on HBsAg isoforms and new therapy approaches that target HBsAg. Still, we believe that the content of these chapters is in the scope of the review’s topic for the following reasons: In the chapter about vaccines, we discuss the presence of different HBsAg isoforms in the vaccines and their influence on the immune response they elicit. The chapter about new antiviral strategies discusses HBsAg isoforms affected by new drugs.
Minor comment
Abstract
- “several HBV-related liver diseases” I suggest to substitute diseases by phases.
This term was substituted by “liver phases and clinical manifestations”.
- I suggest to ease up some affirmations about HBsAg isoforms since many of their functions have not been confirmed so far. For instances, “ Genetic variability of HBsAg components plays a considerable role” I suggest to add a “may”.
We changed the statement as suggested.
- “ for developing generations of HBV”- Omit generations.
The word was omitted.
- “ Initial research has already shown promising results in utilising HBsAg components instead of a whole HBsAg” Substitute whole HBsAg by quantitative HBsAg
The term was substituted.
- “ This review aimed to summarise numerous biological, diagnostic, preventive and therapeutic roles of HBsAg components and emphasise aspects deserving further research.” I suggest modifying this sentences by “this review aimed to summarise the current evidence on the potential usefulness of HBsAg proteins, the limitations for their quantification and aspects deserving further research”
The aim was modified according to suggestion.
Introduction
- “The HBV infection evolves into acute or chronic” Avoid this sentence.
The sentence was omitted.
- I suggest to avoid the previous terms for the HBV phases (i.e. “(“immune tolerant” phase)” or, at least, add before “previously named as”.
The previous terms for HBV phases were omitted.
- “HBsAg-negative phase (“occult HBV infection”). These two terms are not the same; delete the term “occult HBV infection”.
The term was deleted.
- I suggest modifying the last paragraph of this section since it is a copy of the abstract.
The last paragraph was modified.
HBV molecular virology
- “adw, adr, ayw, and 84 ayr. The major antigenic determinant called “a” was unique in all serotypes, while later 85 discovered subdeterminants extended the number of subtypes to 10: ayw1, ayw2, ayw3, 86 ayw4, ayr, adw2, adw3, adwq, adr, and adrq−” This part should be reduced.
This part was reduced.
- I would suggest to create a new section with information regarding HDV.
The paragraph regarding HDV infection was added, and information on HDV in Chapter 3 was modified accordingly.
Impact of HBsAg components on HBV-related diseases
This section refers to HBV mutants, not HBsAg proteins, so I suggest to delete it and to center the manuscript in data on HBsAg proteins.
We believe this section should remain for the reasons we explained in the first paragraph. We added some new information following the suggestions of the other reviewers.
Quantitation of HBsAg isoforms
Add information about the techniques for its performance and the lack of automatized methods.
The information about the techniques for HBsAg isoform quantitation is given in Chapter 5.2, and we added a statement about the need for automatised methods.
6.1. Generations of anti-HBV vaccines based on HBsAg components
I suggest to reduce this section.
The section was reduced.
6.2. New antiviral strategies targeting HBsAg components
This section can also be reduced since it refers to HBsAg, not its proteins.
The section was reduced.
Future directions
“More conclusive evidence can be gathered by future research on the relationship between the variability of surface antigen components and specific clinical entities such as occult hepatitis and HBV reactivation, fulminant hepatitis and HCC development.” Do not mix the potential role of HBsAg proteins and HBV mutants.
We modified the sentence to avoid a potential mix.
Reviewer 3 Report
Comments and Suggestions for Authors
"Hepatitis B surface antigen components: clinical implications, utilisation in diagnosis, prevention and new antiviral strategies" by Lazarevic et al. is a concise and excellently written review covering multiple facets of the Hepatitis B surface antigen and its involvement in pathology, diagnostics and treatment.
The review was informative and an absolute pleasure to read. It is an excellent contribution to the field of HBV research.
I have only very minor grammatical corrections/suggestions for the authors:
Line 198 - replace "defective virus" with "satellite virus"
Line 217 - change "which have several possible consequences like occult HBV infection" to "which may lead to consequences such as occult HBV infection"
Line 231 - "vaccinated children with a strong antibody response to HBsAg could still become HBsAg positive through HBV infection"
Line 269 - change to "Q181*" as * is standard stop codon nomenclature.
Line 275 - " a significant number of cases of chronic hepatitis in the reactivation phase"
Line 320 - change to "carcinogenesis"
Line 321 - remove "The" before HCC
Line 333 - expand VEGFR-2 (VEGF receptor-2), and Akt to Protein Kinase B
Line 335 and 336 - expand NF-kB to "Nuclear factor kappa-light-chain-enhancer of activated B cells" and MAPK to "Mitogen-activated protein kinases"
Line 356 - there is an s in the brackets, please remove.
Line 369 - replace "It participates in" with "It is a target for"
Line 402 - italics for in vitro
Line 440 and 447 - change to "carcinogenesis"
Line 466 - expand NA to "nucleos(t)ide analogs" and PEG-IFN to "pegylated interferon" in the first instance
Line 572 - expand "attachment to the sodium taurocholate cotransporting polypeptide (NTCP) receptor - the cellular receptor for both HBV and HDV"
Comments on the Quality of English Language
Quality of english in the manuscript is excellent - some minor grammatical suggestions have been noted.
Author Response
Response to Reviewer 3
"Hepatitis B surface antigen components: clinical implications, utilisation in diagnosis, prevention and new antiviral strategies" by Lazarevic et al. is a concise and excellently written review covering multiple facets of the Hepatitis B surface antigen and its involvement in pathology, diagnostics and treatment.
The review was informative and an absolute pleasure to read. It is an excellent contribution to the field of HBV research.
I have only very minor grammatical corrections/suggestions for the authors:
Line 198 - replace "defective virus" with "satellite virus"
Line 217 - change "which have several possible consequences like occult HBV infection" to "which may lead to consequences such as occult HBV infection"
Line 231 - "vaccinated children with a strong antibody response to HBsAg could still become HBsAg positive through HBV infection"
Line 269 - change to "Q181*" as * is standard stop codon nomenclature.
Line 275 - " a significant number of cases of chronic hepatitis in the reactivation phase"
Line 320 - change to "carcinogenesis"
Line 321 - remove "The" before HCC
Line 333 - expand VEGFR-2 (VEGF receptor-2), and Akt to Protein Kinase B
Line 335 and 336 - expand NF-kB to "Nuclear factor kappa-light-chain-enhancer of activated B cells" and MAPK to "Mitogen-activated protein kinases"
Line 356 - there is an s in the brackets, please remove.
Line 369 - replace "It participates in" with "It is a target for"
Line 402 - italics for in vitro
Line 440 and 447 - change to "carcinogenesis"
We thank the reviewer for the valuable comments and suggestions. All suggested corrections were made.
Line 466 - expand NA to "nucleos(t)ide analogs" and PEG-IFN to "pegylated interferon" in the first instance
The NA and PEG-IFN abbreviations were explained in the Introduction.
Line 572 - expand "attachment to the sodium taurocholate cotransporting polypeptide (NTCP) receptor - the cellular receptor for both HBV and HDV"
The NTCP abbreviation was explained in Chapter 3 (Biological roles of HBsAg isoforms), and this sentence was expanded by “the cellular receptor for both HBV and HDV”, as suggested.
Comments on the Quality of English Language
Quality of english in the manuscript is excellent - some minor grammatical suggestions have been noted.
Round 2
Reviewer 1 Report
Comments and Suggestions for Authors
No further comments
Comments on the Quality of English Languagemoderate language editing
Author Response
We thank the reviewer for taking the time and effort to read and improve our manuscript with valuable comments. We made an effort to edit the language further according to the suggestions.
Reviewer 2 Report
Comments and Suggestions for Authors
The revised version of the manuscript has improved its quality and it may be accepted for publication after inclusion of minor suggestions:
5.1- Qualitative and quantitative HBsAg detection:
- Avoid the classification of HBV phases by number, and only use the term (e.g. HBeAg-negative infection).
- Line 416: delete the term antibodies.
- Substitute “partial functional cure” by “virological suppression”
Comments on the Quality of English LanguageMinor revision would be adequate.
Author Response
The revised version of the manuscript has improved its quality and it may be accepted for publication after inclusion of minor suggestions:
5.1- Qualitative and quantitative HBsAg detection:
- Avoid the classification of HBV phases by number, and only use the term (e.g. HBeAg-negative infection).
- Line 416: delete the term antibodies.
- Substitute “partial functional cure” by “virological suppression”
We thank the reviewer for taking the time and effort to read and improve our manuscript with valuable comments. We made all the suggested corrections and marked them in the text.